# Single-Loop Multi-Objective Reliability-Based Design Optimization Using Chaos Control Theory and Shifting Vector with Differential Evolution

Raktim Biswas [†] and Deepak Sharma *,[†]

Department of Mechanical Engineering, Indian Institute of Technology Guwahati, Guwahati 781039, Assam, India
* Correspondence: dsharma@iitg.ac.in; Tel.: +91-361-2582661
† These authors contributed equally to this work.

**Abstract:** Multi-objective reliability-based design optimization (MORBDO) is an efficient tool for generating reliable Pareto-optimal (PO) solutions. However, generating such PO solutions requires many function evaluations for reliability analysis, thereby increasing the computational cost. In this paper, a single-loop multi-objective reliability-based design optimization formulation is proposed that approximates reliability analysis using Karush-Kuhn Tucker (KKT) optimality conditions. Further, chaos control theory is used for updating the point that is estimated through KKT conditions for avoiding any convergence issues. In order to generate the reliable point in the feasible region, the proposed formulation also incorporates the shifting vector approach. The proposed MORBDO formulation is solved using differential evolution (DE) that uses a heuristic convergence parameter based on hypervolume indicator for performing different mutation operators. DE incorporating the proposed formulation is tested on two mathematical and one engineering examples. The results demonstrate the generation of a better set of reliable PO solutions using the proposed method over the double-loop variant of multi-objective DE. Moreover, the proposed method requires $6\times$–$377\times$ less functional evaluations than the double-loop-based DE.

**Keywords:** multi-objective reliability-based design optimization; shifting vector approach; reliability analysis; chaos control theory; differential evolution

## 1. Introduction

The design optimization mostly keeps design variables and parameters deterministic. It ignores the fact that uncertainties can arise owing to manufacturing variations, dimensional inaccuracy, boundary conditions, material properties, and improper loading conditions, which can lead to the infeasibility of the solution obtained through deterministic optimization. Therefore, it is necessary to consider these uncertainties in designing the process to maintain safety and the quality of the solution. Reliability-based design optimization (RBDO) [1,2] is a mathematical tool that is used for obtaining such reliable optimal solutions for problems involving uncertainties. It also enables engineers to identify solutions effectively for complex applications in the fields of the automotive, civil, mechanical, and aerospace industries [3,4]. In RBDO, the uncertainties are manifested by converting the deterministic constraints to probabilistic constraints. This is accomplished by applying a probability operator to performance functions or to limit-state functions in the literature. A generalized single-objective RBDO formulation is given in Equation (1).

$$
\begin{aligned}
\text{Minimize} \quad & f(\boldsymbol{\mu_X}), \\
\text{subject to} \quad & P[G_i(\mathbf{X}) \geq 0] \leq P_{f_i}^T = \Phi(-\beta_i^T), \quad i = 1, \ldots, I, \\
& \boldsymbol{\mu_X}^{(L)} \leq \boldsymbol{\mu_X} \leq \boldsymbol{\mu_X}^{(U)},
\end{aligned}
\tag{1}
$$

where $f(\boldsymbol{\mu_X})$ is the objective function, $G_i(\boldsymbol{X})$ is the $i$-th performance/constraint function, and $\boldsymbol{\mu_X}$ is the mean value vector of random variable vector $\boldsymbol{X} \in \mathbb{R}^n$, where $n$ is the number of random design variables. $L$ and $U$ in the superscript of $\boldsymbol{\mu_X}$ represent the lower and upper limits of the vector. $\Phi(\cdot)$ represents the standard normal cumulative distribution function, $\beta_i^T$ is the target reliability index of the $i$-th performance function, and $P[\cdot]$ is the probability operator that represents the failure probability of performance function ($G_i(\boldsymbol{X}) \geq 0$) that should be less than the target failure probability ($P_{f_i}^T$).

Equation (1) demonstrates that solving a single-objective RBDO requires a nested-loop procedure [2], where the outer optimization loop involves the inner-loop for reliability analysis. The reliability analysis can be performed using simulation-based methods [5] and analytical methods [6] on probabilistic performance function to obtain its failure probability. The simulation-based methods show better accuracy with an expense of computational cost [7], such as Monte Carlo simulation (MCS) [5], subset simulation [8], importance sampling [9], and Latin-hypercube sampling [9]. On the other hand, analytical methods are known for their computational efficiency, such as most-probable point (MPP)-based methods, in which the sub-optimization problem is solved for each performance function to obtain their respective MPP. The MPP-based methods can be broadly divided into the performance measurement approach (PMA) [10] and the reliability index approach (RIA) [6]. The optimum solution obtained using PMA and RIA is known as the most probable target point (MPTP) and the most probable failure point (MPFP), respectively. Many advanced methods have been developed to estimate the MPTP and MPFP of performance functions, and they are categorized as double-loop methods, decoupled-loop methods, and single-loop methods.

The classical double-loop methods [11,12] involve a nested optimization loop, where the inner-loop performs reliability analysis and the outer-loop is used for obtaining design solutions. All the random variables are transformed to standard normal variables [13] for performing reliability analysis. Since the nested optimization loop is computationally expensive, the reliability analysis loop (inner-loop) is decoupled and performed separately in decoupled-loop methods [14–17]. Some advanced and efficient reliability-based frameworks were also proposed based on isogeometric analysis [18,19]. The reliability analysis itself is considered as an computationally expensive procedure. Therefore, single-loop methods [20] have been proposed, in which approximate reliability analysis is performed. Different concepts such as Karush-Kuhn Tucker (KKT) conditions and quantile approximation are used to approximate MPTP that can eliminate the reliability analysis loop. The adaptive conjugate single-loop approach (AC-SLA) [21], the enhanced single-loop method (ESM) [22], the chaotic single-loop approach (CSLA) [23], the single-loop shifting vector method (SLShV-CG) [24], the sequential single-loop reliability optimization and confidence analysis method (SROCA) [25], and the approximate single-loop chaos control method (ASLCC) [26] are a few recently developed single-loop methods. Recently, some efficient evolutionary RBDO methods are also proposed to obtain the global reliable solution [27,28].

It has been found that many real-world engineering problems consist of more than one objective, which are conflicting in nature [29], and can also have uncertainties. Evolutionary algorithms are found to be promising for solving deterministic multi-objective optimization problems (MOOPs) because they can generate Pareto-optimal (PO) solutions in one run. However, these evolutionary algorithms need to be modified for generating reliable PO solutions for multi-implemented as a design optimization algorithm, and inverse reliability was performed. objective reliability-based design optimization (MORBDO) problems. To address uncertainty in MORBDO, Deb et al. [3] used a non-dominated sorting genetic algorithm (NSGA-II) [30] for design optimization, and Fast RIA for reliability analysis. A multi-objective differential evolution (MODE) [31] was also Simulation-based techniques are also used for reliability analysis and are coupled with double-loop methods. For example, a radial basis function was used for approximating the responses of the

performance function and was coupled with MCS to implement reliability analysis. NSGA-II was used to obtain PO solutions for solving the multi-objective and multi-case [32] RBDO problem. In another study, MCS and NSGA-II were coupled with entropy weighted grey relational analysis for design optimization [33] to solve the control arm problem. The multi-objective optimization design of the control arm was carried out using the Kriging surrogate model. Sun et al. [34] proposed a radial basis function-based surrogate modeling that was implemented with Latin-hypercube sampling for sensitivity analysis. MCS and multi-objective particle swarm optimization (PSO) were coupled for obtaining the reliable PO solutions. In another study, a multiple response surface method-based artificial neural network was implemented for reliability analysis [35], and a dynamic multi-objective particle swarm optimization algorithm was proposed for obtaining PO solutions. A worst-case scenario was used with fuzzy sets for reliability analysis, and a real-coded population-based incremental learning [36] was implemented with DE for obtaining the PO solutions. A multi-objective robust optimization [37] was proposed, in which the design problems consisted of parametric uncertainties involving both random and interval variables. NSGA-II was implemented to generate robust PO solutions, and MCS was performed to evaluate the impact responses of the mixed uncertainties. Constrained NSGA-II was also implemented to solve the MORBDO problem [38]. It was coupled with the hybrid method using the Kriging surrogate metamodel for reliability analysis.

A time-dependent reliability-based robust design optimization (TRBRDO) problem [39] was solved using NSGA-III [40] and the dimension reduction method. It was developed by constructing an extreme value model using the sparse grid-based stochastic collocation method for time-dependent reliability analysis. A Bayesian multi-objective RBDO [41] was proposed to solve problems involving aleatory and epistemic uncertainties. Multi-objective PSO was implemented for obtaining PO solutions, and Bayesian interference was used for reliability analysis. Another method using nested loop was proposed to solve RBDO problems [42], in which the outer-loop was performed using multi-objective PSO, and the inner-loop was solved using surrogate modeling with MCS sampling. A two-layer nested optimization problem was proposed based on a decoupling strategy. The inter-generation projection genetic algorithm was employed in the inner-loop, and the multi-objective genetic algorithm [43] was implemented at the outer-loop for solving the MORBDO problem. Another multi-objective RBDO [44] was solved by converting it into a single-objective RBDO problem. This was achieved by assigning weights to the objectives based on quantitative analysis and evidence theory. The reliability analysis was estimated using the PMA method.

From the literature, it can be seen that most of the MORBDO methods focus on PMA, RIA, MCS, or surrogate modeling for reliability analysis, and they are based on double-loop or decoupled-loop methods, which make them computationally expensive. Since evolutionary algorithms are population-based methods and require many functional evaluations, a single-loop method for solving MORBDO can improve the computational efficiency. Moreover, single-loop methods that are solved using steepest descent search to estimate MPTP are often stuck with periodic oscillation [26,45] for highly nonlinear functions. This leads to the motivation of this paper, in which a new MORBDO formulation is proposed, based on adaptive multi-objective DE. An adaptive mutation scheme is used for selecting different variants of mutations for exploration in the search space. Both trial and target vectors take part in the MORBDO formulation to estimate the reliable PO solutions. The following are the contributions of the paper.

- A single-loop MORBDO formulation is developed by using a shifting vector approach for achieving feasibility quickly, and by using chaos control theory for estimating MPTP effectively for better convergence.
- An adaptive multi-objective differential evolution is developed by performing two variants of mutation by estimating a heuristic parameter through hypervolume computation.

- The formulation is further developed by incorporating target and trial vectors of differential evolution for better exploration of the search space.

The proposed method is tested on three benchmark examples from the literature. The results are compared with a double-loop variant of multi-objective differential evolution using PMA for reliability analysis.

The organization of the paper is as follows. In Section 2, a brief discussion on multi-objective RBDO, PMA, chaos control method, single-loop method, and shifting vector approach are presented. The proposed single-loop multi-objective reliability-based design optimization method is discussed in Section 3, along with its implementation. The adaptive mutation scheme and the detailed steps of multi-objective differential evolution are also discussed in this section. Numerical examples are solved and discussed in Section 4. Finally, the paper is concluded in Section 5 with a note on future work.

## 2. Preliminaries

### 2.1. Multi-Objective Reliability-Based Design Optimization

A generalized MORBDO formulation can be written as

$$
\begin{aligned}
\text{Minimize} \quad & f_m(\boldsymbol{\mu_X}), & m = 1, \ldots, M, \\
\text{subject to} \quad & P[G_i(\mathbf{X}) \geq 0] \leq P_{f_i}^T = \Phi(-\beta_i^T), & i = 1, \ldots, I, \\
& \boldsymbol{\mu_X^{(L)}} \leq \boldsymbol{\mu_X} \leq \boldsymbol{\mu_X^{(U)}}, \ \mathbf{X}^{(L)} \leq \mathbf{X} \leq \mathbf{X}^{(U)},
\end{aligned}
\tag{2}
$$

where $f_m(\cdot)$ is the $m$-th conflicting objective function that is written using the mean value $(\boldsymbol{\mu_X})$ of the random variable $(\mathbf{X})$. $\mathbf{X}^{(L)}$ and $\mathbf{X}^{(U)}$ are the upper and lower limits on $\mathbf{X}$. Solving Equation (2) generates a set of reliable PO solutions in the design space. The reliability analysis is performed on the probabilistic performance function to estimate the failure probability by solving a multidimensional integral, as given in Equation (3).

$$
P_{f_i} = P[G_i(\mathbf{X}) \geq 0] = \int \cdots \int_{G_i(\mathbf{X}) \geq 0} f_{\mathbf{X}}(\mathbf{X}) d\mathbf{X},
\tag{3}
$$

where $f_{\mathbf{X}}(\mathbf{X})$ is the joint probability density function of $\mathbf{X}$. Solving this multidimensional integral is difficult, and therefore, it is approximated with reliability analysis [7]. The first-order reliability method (FORM) [6] and second-order reliability method (SORM) [46] are analytical methods for reliability analysis. Both FORM and SORM estimate the reliability index $\beta$ that represents the minimum distance from the origin to the performance function in the standard normal space. The reliability index $\beta$ can be obtained by solving a sub-optimization problem, and the reliability (R) can be estimated using $\Phi(\beta)$ (R $= 1 - P_f = 1 - \Phi(-\beta) = \Phi(\beta)$). Due to its computational efficiency and stability in generating a reliable solution, PMA is widely used to solve the sub-optimization problem [47].

### 2.2. Performance Measure Approach (PMA)

PMA estimates the failure probability of performance function $G(\mathbf{X})$ by finding MPTP in the standard normal space ($U$-space). After transforming $G(\mathbf{X})$ to the $U$-space using the Rosenblatt transformation [13], the MPTP can be estimated using the steepest descent direction. When all the random variables are independent, the joint cumulative distribution function (CDF) is calculated via the product of the marginal CDFs. The Rosenblatt transformation is given as

$$
\Phi(u_i) = F_{X_i}(x_i) \implies u_i = \Phi^{-1}(F_{X_i}(x_i)),
\tag{4}
$$

where $F_{X_i}(x_i)$ is the marginal CDF of $X_i$ and $\Phi(\cdot)$ is the CDF of the standard normal random variable. After transforming variables to the standard normal space by using Equation (4), MPTP is calculated by performing the following sub-optimization problem.

$$
\begin{aligned}
\text{Minimize} \quad & G(\mathbf{U}), \\
\text{subject to} \quad & \|\mathbf{U}\| = \beta^T,
\end{aligned}
\tag{5}
$$

where $\mathbf{U}$ is the random variable in the standard normal space, and $\beta^T$ is the target reliability index for the performance function $G(\mathbf{U})$. To efficiently obtain the optimum solution of Equation (5), the advanced mean value algorithm is used and the expression is presented in Equation (6).

$$
\mathbf{U}^{(k+1)} = \beta^T \frac{\nabla G(\mathbf{U})}{\|\nabla G(\mathbf{U})\|}.
\tag{6}
$$

If the performance function value at MPTP is less than or equal to zero, it is satisfied for the given target reliability, as presented in Equation (2).

### 2.3. The Chaos Control Method

It has been observed that PMA performs well for simple nonlinear performance functions, but it fails to converge for highly nonlinear performance functions. To overcome this issue, chaos control theory [45] was proposed based on a stability transformation method [48]. The modification is achieved while updating the iterative point $\mathbf{U}^{(k+1)}$ of Equation (5). The formulation for estimating the iterative point via the chaos control (CC) method is as follows.

$$
\begin{aligned}
\mathbf{U}_{CC}^{(k+1)} &= \mathbf{U}_{CC}^{(k)} + \lambda \mathbf{C}[\mathbf{F}(\mathbf{u}^{(k)}) - \mathbf{U}_{CC}^{(k)}], \\
\mathbf{F}(\mathbf{u}^{(k)}) &= \mathbf{U}^{(k+1)} = \beta^T \frac{\nabla G(\mathbf{U})}{\|\nabla G(\mathbf{U})\|},
\end{aligned}
\tag{7}
$$

where $\mathrm{U}_{CC}^{(k)}$ is the MPTP calculated using CC method in the $k$-th iteration; $\mathbf{C}$ is the involutory matrix with only one element in each row and is assumed as identity matrix $\mathbf{I}$ for simplicity. The matrix $\mathbf{C}$ is usually selected to stabilize the unstable fixed point of the chaotic dynamical system in Equation (7). The chaos control factor $\lambda$ is determined according to the eigenvalues of the original system's Jacobian matrix, and the value is considered within interval $[0, 1]$. When $\lambda$ is considered as one, the formulation of the CC method is similar to Equation (5) and can have the same issue as discussed earlier. Therefore, a small value of $\lambda$ is considered for stable convergence. $\mathbf{F}$ is the vector of the response function that is estimated via nonlinear mapping with respect to the iterative values of $\mathbf{U}^{(k+1)}$, as shown in Equation (7). Although the CC method eliminates the issue of oscillation in the convergence of MPTP, it is considered to be an inefficient process. Therefore, a modified chaos control (MCC) [12] was proposed. The modification is achieved by extending the iterative search to the $\beta$-hypersphere that is at the constraint boundary in the standard normal space. Thus, MPTP is located on the constraint boundary, and convergence is improved by controlling the tangential step size instead of the radial step size, which was the case for the CC method. The formulation of MCC is given as

$$
\begin{aligned}
\tilde{\mathbf{n}}^{(k+1)} &= \mathbf{U}_{CC}^{(k)} + \lambda \mathbf{C}[\mathbf{F}(\mathbf{u}^{(k)}) - \mathbf{U}_{CC}^{(k)}], \\
\mathbf{U}_{MCC}^{(k+1)} &= \beta^T \frac{\tilde{\mathbf{n}}^{(k+1)}}{\|\tilde{\mathbf{n}}^{(k+1)}\|},
\end{aligned}
\tag{8}
$$

where $\tilde{\mathbf{n}}^k$ is the modified search direction updated using $\mathbf{U}_{CC}^{(k+1)}$ of Equation (7). $\mathbf{U}_{MCC}^{(k+1)}$ is the MPTP evaluated using the MCC method.

### 2.4. Single-Loop Method

The single-loop method (SLM) [20] has been proposed to approximate the reliability analysis of the double-loop method, and establish an equivalent deterministic performance function that is computationally efficient. The approximate MPTP is estimated by using the KKT optimality conditions of Equation (5), and is given in Equation (9).

$$\nabla G(\mathbf{U}) - \hat{\lambda} \nabla H(\mathbf{U}) = 0, \tag{9}$$

where $\hat{\lambda}$ is the Lagrange multiplier, and $H(\mathbf{U}) = \|\mathbf{U}\|^2 - \beta_i^{T^2}$ after squaring both sides of the equality constraint of Equation (5). Using Equation (9) and $\nabla H(\mathbf{U}) = 2\mathbf{U}$ yields $\nabla G(\mathbf{U}) - 2\mathbf{U}\hat{\lambda} = 0$. After simplification, $\mathbf{U}$ can be written as $\frac{\nabla G(\mathbf{U})}{2\hat{\lambda}}$, and multiplying it with $\|\nabla G(\mathbf{U})\|$ in the numerator and denominator, and further simplifying, we obtain

$$\mathbf{U} = \frac{\|\nabla G(\mathbf{U})\|}{2\hat{\lambda}} \frac{\nabla G(\mathbf{U})}{\|\nabla G(\mathbf{U})\|} = \beta^T \boldsymbol{\alpha}, \tag{10}$$

where $\boldsymbol{\alpha} = \frac{\nabla G(\mathbf{U})}{\|\nabla G(\mathbf{U})\|}$ is the unit gradient direction, and $\beta^T = \frac{\|\nabla G(\mathbf{U})\|}{2\hat{\lambda}}$ is a constant at the optimal solution $\mathbf{U}^*$. The gradient is calculated in $U$-space and the random design variables lie in the $X$-space. Therefore, the transformation from $X$-space to $U$-space is used for the evaluation of approximate MPTP, using the following relationship.

$$\mathbf{X} = \boldsymbol{\mu_X} + \sigma_\mathbf{X} \mathbf{U}, \tag{11}$$

where $\sigma_\mathbf{X}$ is the standard deviation of $\mathbf{X}$. Substituting $\mathbf{U}$ from Equation (10) in Equation (11) and using the chain rule, we obtain MPTP in the $X$-space as

$$\mathbf{X}_{MPTP} = \boldsymbol{\mu_X} + \sigma_\mathbf{X} \beta \boldsymbol{\alpha} = \boldsymbol{\mu_X} + \sigma_\mathbf{X} \beta^T \frac{\sigma_\mathbf{X} \nabla_\mathbf{X} G(\mathbf{X})}{\|\sigma_\mathbf{X} \nabla_\mathbf{X} G(\mathbf{X})\|}, \tag{12}$$

where $\mathbf{X}_{MPTP}$ is the MPTP of the performance function $G(\mathbf{X})$.

### 2.5. Shifting Vector Approach

The concept of the shifting vector $(\mathbf{S}_i^{(k)})$ has been proposed [14] to decouple the double-loop structure of the RBDO problem. It separates the optimization and reliability analysis loop and performs it sequentially in the sequential optimization and reliability assessment (SORA) [14] method. Using this process, the computational efficiency of SORA has been improved as compared to the double-loop method. The concept of the shifting vector is used to shift the violated performance function towards the feasible direction. It is given as

$$\mathbf{S}_i^{(k)} = \boldsymbol{\mu_X}^{(k-1)} - \mathbf{X}_{i,MPTP}^{(k-1)} \tag{13}$$

where $(\mathbf{S}_i^{(k)})$ is the shifting vector at the $k$-th iteration, $\mathbf{X}_{i,MPTP}^{(k-1)}$ is the MPTP for the $i$-th constraint, and $\boldsymbol{\mu_X}^{(k-1)}$ is the mean of the random variable $\mathbf{X}$ in the $(k-1)$-th iteration. Figure 1 shows the schematic diagram of the shifted constraint based on the MPTP. It can be seen that $(\mathbf{S}_i^{(1)})$ is estimated based on $\mathbf{X}_{i,MPTP}^{(1)}$ and $\boldsymbol{\mu_X}^{(1)}$, and the shifted constraint is evaluated at $\boldsymbol{\mu_X}^{(1)} - (\mathbf{S}_i^{(1)})$ until the reliability of the constraint is achieved. Here, the shifting vector $(\mathbf{S}_i^{(k)})$ is generated via an iterative process that helps to estimate the feasibility of the performance function until its reliability is satisfied.

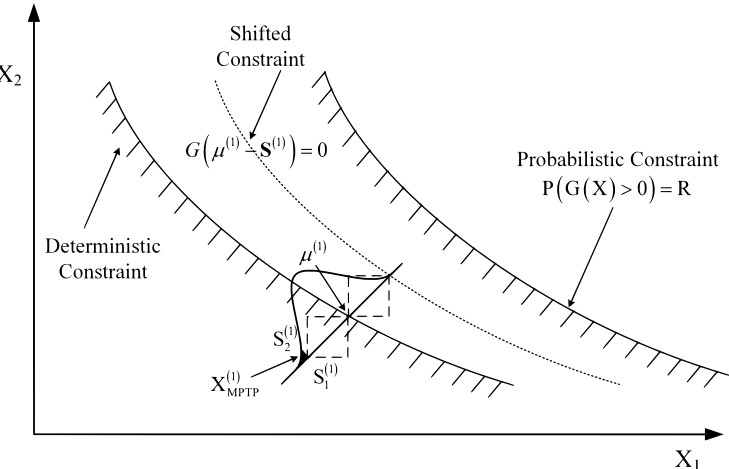

**Figure 1.** Shifting vector approach.

## 3. The Proposed Method and Its Implementation

### 3.1. Single-Loop MORBDO Formulation Using Chaos Control and the Shifting Vector Approach

The single-loop MORBDO formulation can be written using the approximate MPTP given in Equation (12) as

$$
\begin{aligned}
&\text{Min. } f_m(\boldsymbol{\mu_X}), && m = 1, \dots, M, \\
&\text{s.t.: } G_i(\mathbf{X}_{i,MPTP}^{(k)}) \leq 0, && i = 1, \dots, I, \\
&\text{where } \mathbf{X}_{i,MPTP}^{(k)} = \boldsymbol{\mu_X}^{(k)} + \beta_i^T \sigma_{\mathbf{X}} \boldsymbol{\alpha}_{i,\mathbf{X}}^{(k)}, \\
&\boldsymbol{\alpha}_{i,\mathbf{X}}^{(k)} = \frac{\sigma_{\mathbf{X}} \nabla G_{i,\mathbf{X}}(\mathbf{X}_{i,MPTP}^{(k-1)})}{\|\sigma_{\mathbf{X}} \nabla G_{i,\mathbf{X}}(\mathbf{X}_{i,MPTP}^{(k-1)})\|}, \\
&\boldsymbol{\mu_X}^{(L)} \leq \boldsymbol{\mu_X} \leq \boldsymbol{\mu_X}^{(U)},
\end{aligned}
\tag{14}
$$

where $\mathbf{X}_{i,MPTP}^{(k)}$ is the approximate MPTP of the '$i$' performance function at the $k$-th iteration, and $\boldsymbol{\alpha}_{i,\mathbf{X}}^{(k)}$ is the unit gradient vector of the performance function '$i$' with respect to random variable (**X**). In Equation (14), the probabilistic performance functions of Equation (2) are converted into deterministic performance functions, which eliminate the MPTP search of the inner-loop at every iteration. Thus, the computational efficiency can be improved significantly. It is to be noted that the steepest descent search is used to evaluate the approximate MPTP, which has a tendency to oscillate during convergence [45].

In the proposed formulation, chaos control theory replaces the steepest descent search for approximating MPTP. The concept of the shifting vector approach is incorporated to formulate a novel single-loop MORBDO formulation, as shown in Equation (15).

$$
\begin{aligned}
&\text{Min. } f_m(\boldsymbol{\mu_X}), && i = 1, \dots, M, \\
&\text{s.t.: } G_i(\boldsymbol{\Psi}^{(k)}) \leq 0, && i = 1, \dots, I, \\
&\text{where } \boldsymbol{\Psi}^{(k)} = 
\begin{cases}
\mathbf{X}_{i,MPTP}^{(k)}, & \forall \text{ target vectors}, \\
\boldsymbol{\mu_U}^{(k+1)} - \mathbf{S}_i^{(k+1)}, & \forall \text{ trial vectors},
\end{cases} \\
&\mathbf{S}_i^{(k+1)} = \boldsymbol{\mu_X}^{(k)} - \mathbf{X}_{i,MPTP}^{(k)}, \\
&\mathbf{X}_{i,MPTP}^{(k)} = \mathrm{T}^{-1}(\mathbf{U}) = \boldsymbol{\mu_X}^{(k)} + \sigma_{\mathbf{X}} \mathbf{U}_{i,SLCC}^{(k)}, \\
&\boldsymbol{\mu_X}^{(L)} \leq \boldsymbol{\mu_X} \leq \boldsymbol{\mu_X}^{(U)},
\end{aligned}
\tag{15}
$$

where $\mathbf{U}^{(k)}_{i,SLCC}$ is the approximate MPTP in the $U$-space that is estimated using the MCC method. $\boldsymbol{\mu}^{(k+1)}_{\mathbf{U}}$ is the trial vector of differential evolution in the $U$-space in the $(k+1)$-th iteration. In the proposed formulation, the performance function $G_i(\boldsymbol{\Psi}^{(k)})$ includes both $\mathbf{X}^{(k)}_{i,MPTP}$ and $(\boldsymbol{\mu}^{(k+1)}_{\mathbf{U}} - \mathbf{S}^{(k+1)}_i)$, which are used for evaluating the performance function for each target vector and trial vector, respectively. The vector $(\boldsymbol{\mu}^{(k+1)}_{\mathbf{U}} - \mathbf{S}^{(k+1)}_i)$ shifts the violated performance function towards a feasible direction for the population of trial vectors. $\mathbf{U}^{(k)}_{i,SLCC}$ in the standard normal space is given in Equation (16).

$$\mathbf{U}^{(k)}_{i,SLCC} = \beta^T_i \frac{\mathbf{U}^{(k-1)}_i + \lambda^{(k)}_i \mathbf{C}[\mathbf{U}^{(k)}_i - \mathbf{U}^{(k-1)}_i]}{\|\mathbf{U}^{(k-1)}_i + \lambda^{(k)}_i \mathbf{C}[\mathbf{U}^{(k)}_i - \mathbf{U}^{(k-1)}_i]\|}, \tag{16}$$

where $\mathbf{U}^{(k)}_i$ and $\mathbf{U}^{(k-1)}_i$ are the MPTPs estimated for the $i$-th constraint in the $k$-th and $(k-1)$-th generations, respectively. The value of $\mathbf{U}^{(k)}_{i,SLCC}$ is calculated after the transformation, as given in Equation (17).

$$\mathbf{U}^{(k)}_i = \mathrm{T}(\mathbf{X}^{(k)}_i) = (\mathbf{X}^{(k)}_i - \boldsymbol{\mu}_{\mathbf{X}})/\sigma_X. \tag{17}$$

The proposed single-loop MORBDO formulation given in Equation (15) is developed based on a single-loop methodology that eliminates the integrated reliability analysis involved in double-loop formulation, as given in Equation (2). The approximated formulation for reliability analysis is established through KKT optimality conditions, where the search direction is calculated by using modified chaos control theory. Furthermore, the shifting vector is integrated with the single-loop MORBDO formulation that uniquely involves the target and trial vectors of differential evolution.

### 3.2. Multi-Objective Differential Evolution with Adaptive Mutation Scheme

Differential evolution (DE) [49] is a population-based meta-heuristic algorithm that works with a set of vectors and optimizes an optimization problem by iteratively improving each vector based on an evolutionary process. It explores the design space by maintaining a population of vectors and creating new vectors by combining existing ones. It starts with a random generation of vectors, which are referred to as target vectors, $\boldsymbol{\mu}^{(k)}_{\mathbf{X}}(t)$, in which $t$ represents the $t$-th target vector, and $k$ represents the $k$-th generation counter. Since DE is used for solving the MORBDO problem, the notation for vector is kept the same as the mean value of the random variable. Each target vector $(\boldsymbol{\mu}^{(k)}_{\mathbf{X}}(t))$ is transformed to the mutant vector $(\boldsymbol{\mu}^{(k+1)}_{\mathbf{V}}(t))$ using the randomly chosen vectors $(\boldsymbol{\mu}^{(k)}_{\mathbf{r}_1}(t))$, $(\boldsymbol{\mu}^{(k)}_{\mathbf{r}_2}(t))$ and $(\boldsymbol{\mu}^{(k)}_{\mathbf{r}_3}(t))$. In this paper, an adaptive mutation scheme is used, in which the mutation vector $(\boldsymbol{\mu}^{(k+1)}_{\mathbf{V}}(t))$ is generated, either by using a random vector or the best vector. The scheme for generating $(\boldsymbol{\mu}^{(k+1)}_{\mathbf{V}}(t))$ is given in Equation (18).

$$\boldsymbol{\mu}^{(k+1)}_{\mathbf{V}}(t) = \begin{cases} \boldsymbol{\mu}^{(k)}_{\mathbf{r}1}(t) + \hat{F} \times (\boldsymbol{\mu}^{(k)}_{\mathbf{r}2}(t) - \boldsymbol{\mu}^{(k)}_{\mathbf{r}3}(t)), & \zeta > \epsilon, \\[2mm] \boldsymbol{\mu}^{(k)}_{\mathbf{best}}(t) + \hat{F} \times (\boldsymbol{\mu}^{(k)}_{\mathbf{r}2}(t) - \boldsymbol{\mu}^{(k)}_{\mathbf{r}3}(t)), & \text{otherwise,} \end{cases} \tag{18}$$

where $\mathbf{r_1} \neq \mathbf{r_2} \neq \mathbf{r_3}$ are the three randomly chosen vectors from the current population, and $\hat{F}$ is the scaling factor. The variant "DE/rand/bin/1" is found to be effective in exploring the search space during the initial generations because the mutant vector is generated a using random vector. When DE starts converging towards the Pareto-optimal front, the "DE/best/bin/1" variant replacing $\boldsymbol{\mu}^{(k)}_{\mathbf{r}_1}(t)$ to $\boldsymbol{\mu}^{(k)}_{\mathbf{best}}(t)$ can improve the convergence. The $\boldsymbol{\mu}^{(k)}_{\mathbf{best}}(t)$ vector for each target vector is found by calculating the Euclidean distance of the $t$-th target vector with respect to all non-dominated target vectors in the objective

space. The closest non-dominated target vector is selected as $\boldsymbol{\mu}_{\mathbf{best}}^{(k)}(t)$ for the $t$-th target vector. Since both the variants have their own merits, a heuristic convergence parameter ($\zeta$) is proposed that can help DE to use either of these variants, depending on the user-defined parameter $\epsilon$. The parameter $\zeta$ is calculated using the hypervolume (HV) performance indicator [50] that is given as

$$\zeta = I_H^{(k)} - I_H^{(k-1)}, \tag{19}$$

where $I_H^{(k)}$ and $I_H^{(k-1)}$ are the hypervolume calculated with respect to the non-dominated target vectors in the $(k)$ and $(k-1)$ generations. It is noted that the non-dominated target vectors in the $(k-1)$ and $(k)$ generations are normalized together for estimating the hypervolume with respect to the dominated point. Thereafter, the trial vector ($\boldsymbol{\mu}_{\mathbf{U}}^{(k+1)}(t)$) is created for each target vector ($\boldsymbol{\mu}_{\mathbf{X}}^{(k)}(t)$), which is given as

$$\boldsymbol{\mu}_{\mathbf{U}}^{(k+1)}(t_j) = \begin{cases} \boldsymbol{\mu}_{\mathbf{V}}^{(k+1)}(t_j) & \text{if } r \leq p_c \text{ or } j = rnbr(i), \\ \boldsymbol{\mu}_{\mathbf{X}}^{(k)}(t_j) & \text{if } r > p_c \text{ and } j \neq rnbr(i), \end{cases} \tag{20}$$

where subscript $j$ with $t$ in $\boldsymbol{\mu}_{\mathbf{X}}^{(k)}(t_j)$, $\boldsymbol{\mu}_{\mathbf{V}}^{(k+1)}(t_j)$, and $\boldsymbol{\mu}_{\mathbf{U}}^{(k+1)}(t_j)$ represent the $j$-th component of the target, mutant, and trial vectors, respectively. $r$ is a random number between 0 and 1, $p_c$ is the crossover rate, and $rnbr(i)$ is a randomly chosen index $\in \{1, 2, \ldots, n\}$, which ensures that $\boldsymbol{\mu}_{\mathbf{U}}^{(k+1)}(t_j)$ obtains at least one component from $\boldsymbol{\mu}_{\mathbf{V}}^{(k+1)}(t_j)$. Thereafter, all target vectors and trial vectors are combined ($\boldsymbol{\mu}_{\mathbf{X}}^{(k)} \bigcup \boldsymbol{\mu}_{\mathbf{U}}^{(k+1)}$) to find the rank of the combined population using the non-dominated sorting [30] of NSGA-II. The crowding distance is also calculated for maintaining the diversity for the selection of the next generation of target vectors. The best $N$ target vectors for the next generation are selected by using the environmental selection scheme of NSGA-II [30]. Multi-objective DE is terminated if the generation counter ($k$) is more than the total number of generations ($K$). Otherwise, the generation loop continues till the termination condition becomes satisfied.

*3.3. Steps for Implementation*

In this section, the steps for implementing DE with an adaptive mutation scheme for the proposed MORBDO formulation are presented, which are as follows.

1  **Input**: population size ($N$), number of variables ($n$), total number of generations ($K$), scaling factor ($\hat{F}$), probability of crossover ($p_c$), standard deviation ($\sigma$) for random variables, and target reliability index for constraints ($\beta^T$), generation counter ($k = 1$).

2  Initialize random population ($P(k)$) that comprises target vectors ($\boldsymbol{\mu}_{\mathbf{X}}^{(k)}$).

3  For each target vector ($\boldsymbol{\mu}_{\mathbf{X}}^{(k)}(t)$) of ($P(k)$):

   3.1  Calculate the objective function values, $f_m(\boldsymbol{\mu}_{\mathbf{X}}^{(k)}(t))$.

   3.2  Calculate MPTP for each performance function ($i$) using Equations (15) and (16), and estimate shifting vector $\mathbf{S}_{i,\boldsymbol{\mu}_{\mathbf{X}}}^{(k+1)} = \boldsymbol{\mu}_{\mathbf{X}}^{(k)} - \mathbf{X}_{i,MPTP}^{(k)}$.

   3.3  Calculate the constraint violation of each performance function using the MPTP that is estimated through the chaos control theory given in Equation (15).

4  If ($k > K$), terminate. Otherwise, continue to Step 5.

5  Generate mutant vectors ($\boldsymbol{\mu}_{\mathbf{V}}^{(k+1)}$) using the scheme given in Equation (18).

6  Generate trial vectors ($\boldsymbol{\mu}_{\mathbf{U}}^{(k+1)}$), as given in Equation (20).

7  For each trial vector:

   7.1  Calculate the objective function, $f_m(\boldsymbol{\mu}_{\mathbf{U}}^{(k+1)}(t))$.

7.2 Calculate MPTP $(\hat{\mathbf{X}}_{i,MPTP}^{(k+1)})$ and shifting vector for each performance function $(i)$ using Equations (15) and (16), and $\mathbf{S}_{i,\mu_{\mathbf{U}}}^{(k+2)} = \mu_{\mathbf{U}}^{(k+1)} - \hat{\mathbf{X}}_{i,MPTP}^{(k+1)}$, and estimate the constraint violation of $G_i(\mu_{\mathbf{U}}^{(k+1)} - \mathbf{S}_{i,\mu_{\mathbf{X}}}^{(k+1)})$.

7.3 Calculate the constraint violation of each performance function using the MPTP that is estimated through chaos control theory as given in Equation (15).

8  Combine target and trial vectors, and perform non-dominated sorting and estimate the crowding distance.

9  Update target vectors $(\mu_{\mathbf{X}}^{(k+1)})$ for the next generation using the environmental selection of NSGA-II. It should be noted that the corresponding MPTPs and shifting vector are stored in Step 7 for utilizing them in the next generation. Set $k = k + 1$ and go to Step 4.

## 4. Numerical Examples

In this section, three mathematical examples and one engineering example are solved to demonstrate the performance of the proposed method. All the examples consist of two objective functions, along with the nonlinear performance functions. The proposed method is abbreviated as SLMDE since it is developed via a single-loop method using multi-objective DE. The results of SLMDE are compared with double-loop multi-objective differential evolution (DLMDE). It is noted that PMA is used with DLMDE for reliability analysis. The reliable PO solutions are generated via both methods for different values of the target reliability index $(\beta^T)$. HV performance indicator values and number of function evaluations are used to compare the outcome. Both the methods are run 30 times with different initial populations. The standard deviation (SD) is also evaluated to see the dispersion of HV values. The Wilcoxon signed-rank test at a 5% significance level is also used to determine the difference for the statistical significance between SLMDE and DLMDE. The parameters of SLMDE and DLMDE are as follows: the scaling factor $(\hat{F})$ is taken as 0.3, the crossover probability $(p_c)$ is 0.9, the population size $(N)$ is 200, and the total number of generations $(K)$ is 100 for the first example, 250 for the second example, and 200 for the car side impact example. The chaos control factor $(\lambda)$ is considered as 0.2 [26]. The user-defined parameter $(\epsilon)$ in Equation (18) is considered as $10^{-3}$. The MATLAB R2016b platform is used for developing both methods.

### 4.1. Example 1

The first MORBDO example [3] consists of two objectives that are developed using two independent random normal variables with a standard deviation of 0.03. The example is subjected to two linear performance functions that are shown in Equation (21).

$$
\begin{aligned}
\text{min: } & f_1(\mu_{\mathbf{X}}) = \mu_{x_1}, \\
\text{min: } & f_2(\mu_{\mathbf{X}}) = \frac{1 + \mu_{x_2}}{\mu_{x_1}}, \\
\text{s.t.: } & P[G_i(\mathbf{X}) > 0] \leq \phi(-\beta_i^T), \quad i = 1, 2, \\
& G_1(\mathbf{X}) = x_2 + 9x_1 - 6, \\
& G_2(\mathbf{X}) = -x_2 + 9x_1 - 1, \\
& 0.1 \leq \mu_{x_1} \leq 1, \ 0 \leq \mu_{x_1} \leq 5.
\end{aligned}
\tag{21}
$$

Table 1 presents the best, median, and worst values of HV obtained via SLMDE and DLMDE. SLMDE has converged to better values of HV for different $\beta^T$ values. This indicates that SLMDE generates a better set of PO solutions for the given example. It is to be noted that for a larger value of $\beta^T$, the HV value becomes reduced, as compared to the lower $\beta^T$ value. This is because a larger value of $\beta^T$ signifies a high degree of reliability that makes the obtained PO solutions more conservative and pushes them away from the deterministic PO front inside the feasible region.

**Table 1.** Best, median, and worst HV values obtained by both methods are presented for Example 1 for different values of $\beta^T$. The best performances are highlighted in bold font.

| $\beta^T$ | SLMDE | DLMDE | $\beta^T$ | SLMDE | DLMDE | $\beta^T$ | SLMDE | DLMDE |
|---|---|---|---|---|---|---|---|---|
| | **0.8100** | 0.8067 | | **0.7998** | 0.7754 | | **0.7908** | 0.7422 |
| 1.0 | **0.8075**$^+$ | 0.8048 | 2.0 | **0.7982**$^+$ | 0.7739 | 3.0 | **0.7879**$^+$ | 0.7411 |
| | **0.8065** | 0.7772 | | **0.7971** | 0.7550 | | **0.7869** | 0.7247 |
| SD | $\mathbf{9 \times 10^{-4}}$ | 0.0063 | SD | $\mathbf{6 \times 10^{-4}}$ | 0.0042 | SD | $\mathbf{7 \times 10^{-4}}$ | 0.0054 |

Figure 2 demonstrates the PO solutions obtained by both methods for different values of $\beta^T$. The reliable PO solutions shown in Figure 2a,b correspond to the median HV values from Table 1. It can be seen that for larger values of $\beta^T$, the PO solutions become conservative and move inside the feasible region. The same figure also demonstrates that some solutions coincide with the deterministic PO front that is located at the bottom right. This is because for those solutions, the target reliability is satisfied for the performance function $G_1(\mathbf{x})$.

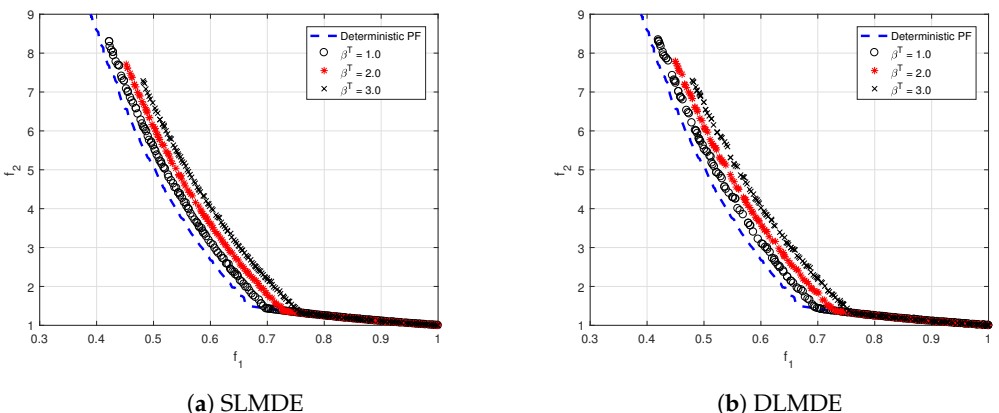

(**a**) SLMDE                    (**b**) DLMDE

**Figure 2.** The obtained PO solutions by both methods for example 1 for different $\beta^T$ values.

The computational efficiencies of both methods are measured with the help of a number of function evaluations that are presented in Table 2. It can be seen that the proposed method requires 202,000 function evaluations, which is only 14.85% of DLMDE. This is because SLMDE is based on a single-loop method, where the reliability of the performance function is estimated using KKT optimality conditions. On the other hand, DLMDE performs PMA for reliability estimation, which requires many function evaluations. Since the number of iterations for PMA is kept fixed, the number of function evaluations is the same for DLMDE with different values of $\beta^T$.

**Table 2.** Number of function evaluations required by both methods for example 1.

| $\beta^T$ | SLMDE | DLMDE |
|---|---|---|
| 1.0 | 202,000 | 1,360,000 |
| 2.0 | 202,000 | 1,360,000 |
| 3.0 | 202,000 | 1,360,000 |

The Wilcoxon test results are shown in the same table with symbols $(+, =, -)$. The symbol '+' suggests a significantly better performance of SLMDE over DLMDE. Other symbols '−' and '=' suggest a significantly bad performance and an equivalent performance of SLMDE over DLMDE, respectively. It can be seen from the table that SLMDE shows a significantly better performance over DLMDE.

The progress of HV and heuristic convergence parameter $\zeta$ with respect to iterations are shown in Figure 3. It can be seen that there are some initial fluctuations in both HV and $\zeta$, which subsidise after 10 generations and stabilize after 50 generations.

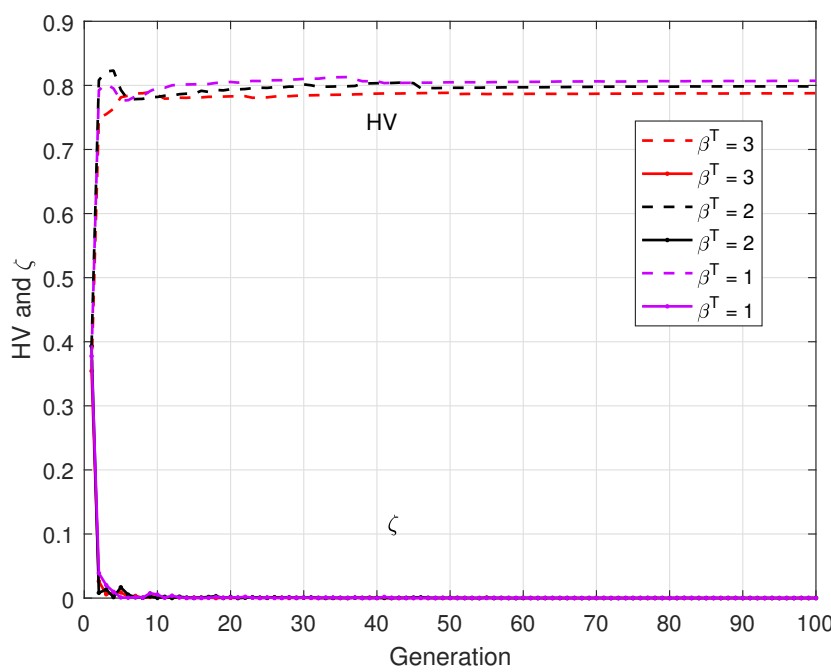

**Figure 3.** Progress of hypervolume and $\zeta$ of SLMDE with respect to number of generations for example 1.

*4.2. Example 2*

The second example [51] consists of two objective functions which are highly nonlinear. It has four linear and two nonlinear performance functions that are developed using two independent random normal variables, each with a standard deviation of 0.3. The RBDO formulation of this example is given in Equation (22).

$$\text{min: } f_1(\boldsymbol{\mu_X}) = -[25(\mu_{x_1} - 2)^2 + (\mu_{x_2} - 2)^2 + (\mu_{x_3} - 1)^2 + (\mu_{x_4} - 4)^2 + (\mu_{x_5} - 1)^2],$$

$$\text{min: } f_2(\boldsymbol{\mu_X}) = [\mu_{x_1}^2 + \mu_{x_2}^2 + \mu_{x_3}^2 + \mu_{x_4}^2 + \mu_{x_5}^2 + \mu_{x_6}^2],$$

$$\text{s.t.: } P[G_i(\mathbf{X}) > 0] \leq \phi(-\beta_i^T), \quad i = 1, \ldots, 6$$

$$G_1(\mathbf{X}) = x_1 + x_2 - 2,$$

$$G_2(\mathbf{X}) = 6 - x_1 - x_2,$$

$$G_3(\mathbf{X}) = 2 - x_2 + x_1,$$

$$\quad (22)$$

$$G_4(\mathbf{X}) = 2 - x_1 + 3x_2,$$

$$G_5(\mathbf{X}) = 4 - (x_3 - 3)^2 - x_4,$$

$$G_6(\mathbf{X}) = (x_5 - 3)^2 + x_6 - 4,$$

$$0 \leq \mu_{x_1}, \mu_{x_2}, \mu_{x_6} \leq 10, \ 1 \leq \mu_{x_3}, \mu_{x_5} \leq 5, \ 0 \leq \mu_{x_4} \leq 6.$$

Table 3 presents the statistical values of HV obtained via both methods. In can be seen that SLMDE has converged to better values of HV for different $\beta^T$ values. This indicates that SLMDE generates a better set of PO solutions for this given example. In this case, a similar observation can also be made where for larger values of $\beta^T$, the HV values becomes reduced. This is due to the fact that larger values of $\beta^T$ signify a larger degree of reliability,

which leads to the generation of conservative PO solutions. The Wilcoxon test results are shown in the same table with symbols $(+, =, -)$. It can be seen from the table that SLMDE shows a significantly better performance over DLMDE.

Figure 4 shows the reliable PO solutions generated in the run, corresponding to a median HV value from Table 3. It can be seen that for larger $\beta^T$, PO solutions move inside the feasible region and away from the deterministic PO front. The spread of solutions is less in the case of SLMDE for $\beta^T = 1.0$. The solutions are nicely distributed in the case of DLMDE. The shift of the solutions is more for larger values of $\beta^T$, which leads to smaller values of HV that can be seen from Table 3.

**Table 3.** Best, median, and worst HV values obtained by both methods are presented for Example 2 for different values of $\beta^T$. The best performances are highlighted in bold font.

| $\beta^T$ | SLMDE | DLMDE | $\beta^T$ | SLMDE | DLMDE | $\beta^T$ | SLMDE | DLMDE |
|---|---|---|---|---|---|---|---|---|
| | **0.9118** | 0.9017 | | **0.6234** | 0.6119 | | **0.4027** | 0.3929 |
| 1.0 | **0.9028**$^+$ | 0.8906 | 2.0 | **0.6181**$^+$ | 0.6025 | 3.0 | **0.3945**$^+$ | 0.3777 |
| | **0.8853** | 0.8293 | | **0.5999** | 0.5953 | | **0.3776** | 0.3680 |
| SD | **0.0065** | 0.0153 | SD | 0.0065 | **0.0056** | SD | **0.0074** | 0.0080 |

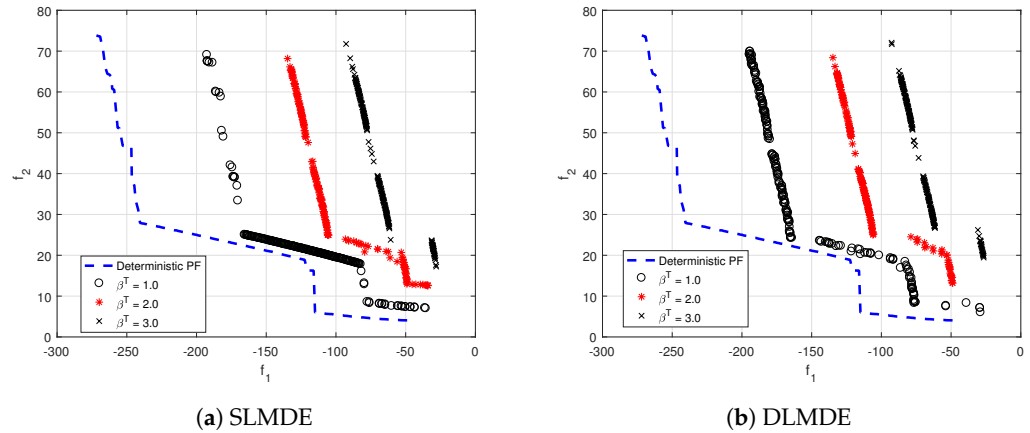

(**a**) SLMDE          (**b**) DLMDE

**Figure 4.** The PO solutions obtained via both methods for example 2, for different $\beta^T$ values.

Table 4 presents the computational efficiency of both methods. The proposed method only requires 3,915,600 function evaluations, which is only 3.5–2.7% of DLMDE. It suggests that DLMDE needs many function evaluations because PMA is performed for reliability estimation.

**Table 4.** Number of function evaluations required by both methods for Example 2.

| $\beta^T$ | SLMDE | DLMDE |
|---|---|---|
| 1.0 | 3,915,600 | 111,424,992 |
| 2.0 | 3,915,600 | 132,344,016 |
| 3.0 | 3,915,600 | 145,249,728 |

Figure 5 shows the progress of HV and $\zeta$ with respect to the number of generations. It can be seen that there are fluctuations for all values of $\beta^T$ until the termination criterion is achieved. The initial fluctuations can also be observed for $\zeta$, which subsidize after 150 generations.

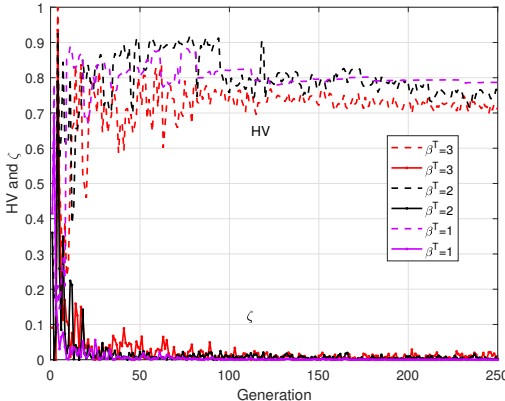

**Figure 5.** Progress of hypervolume and $\zeta$ of SLMDE with respect to the number of generations for example 2.

*4.3. Car Side Impact Example*

The car side impact [3] example is considered as an engineering RBDO example, which is formulated by using 2 objectives and 10 performance functions. It consists of 11 random design variables that are normally distributed and that are grouped into random variables $(x_1, \ldots, x_7)$ and random parameters $(x_8, \ldots, x_{11})$. The details of the variables with their standard deviation values are given in Table 5. The RBDO formulation is presented in Equation (23). The mathematical expressions for each function are given in Table 6.

$$
\begin{aligned}
&\text{min: } f_1(\boldsymbol{\mu_X}) \equiv \text{Structural weight,} \\
&\text{min: } f_2(\boldsymbol{\mu_X}) \equiv \text{Average rib deflection,} \\
&\text{s.t.: } P[G_i(\boldsymbol{X}) > 0] \leq \phi(-\beta_i^T), \quad i = 1, \ldots, 10 \\
&\quad G_1(\boldsymbol{X}) = \text{Abdomen load} \leq 1\text{KN,} \\
&\quad G_2(\boldsymbol{X}) = V * Cupper \leq 0.32\text{m/s,} \\
&\quad G_3(\boldsymbol{X}) = V * Cmiddle \leq 0.32\text{m/s,} \\
&\quad G_4(\boldsymbol{X}) = V * Clower \leq 0.32\text{m/s,} \\
&\quad G_5(\boldsymbol{X}) = \text{Upper rib deflection} \leq 32\text{mm,} \\
&\quad G_6(\boldsymbol{X}) = \text{Middle rib deflection} \leq 32\text{mm,} \\
&\quad G_7(\boldsymbol{X}) = \text{Lower rib deflection} \leq 32\text{mm,} \\
&\quad G_8(\boldsymbol{X}) = \text{Pubic force} \leq 4\text{KN,} \\
&\quad G_9(\boldsymbol{X}) = \text{Velocity of V-Pillar} \leq 9.9\text{mm/ms,} \\
&\quad G_{10}(\boldsymbol{X}) = \text{Front door velocity of V-Pillar} \leq 15.7\text{mm/ms,} \\
&\quad 0.5 \leq \mu_{x_1}, \mu_{x_3}, \mu_{x_4} \leq 1.5, \ 0.45 \leq \mu_{x_2} \leq 1.35, \ 0.875 \leq \mu_{x_5} \leq 2.625, \\
&\quad 0.4 \leq \mu_{x_6} \leq 1.2, \ 0.4 \leq \mu_{x_7} \leq 1.2, 0.192 \leq \mu_{x_8}, \mu_{x_9} \leq 0.75.
\end{aligned}
\tag{23}
$$

The statistical values of the HV values obtained from both methods with respect to different $\beta^T$ are presented in Table 7. The proposed method has converged to larger values of HV for all $\beta^T$ values. This signifies a better distribution of PO solutions of SLMDE as compared to DLMDE. The observation of reducing HV values with larger $\beta^T$ values remains the same. The Wilcoxon test results are shown in the same table with symbols $(+, =, -)$. It can be seen from the table that SLMDE shows significantly better, bad, and equivalent performances over DLMDE for $\beta^T = 1, 2$ and 3, respectively.

The obtained reliable PO solutions for both methods are shown in Figure 6. As observed with previous examples, for larger values of $\beta^T$, the PO solutions start moving away from the deterministic PO front inside the feasible region.

Table 8 presents the computational efficiency of both methods. In this example, SLMDE requires only 4,623,000, the number of function evaluations, which is only 0.3–0.26% that of

DLMDE. Since SLMDE performs an approximate reliability estimation by using the KKT optimality conditions, it saves many function evaluations compared to DLMDE. Figure 7 shows a similar progress for HV and $\zeta$ with respect to the number of generations. There are initial fluctuations for all values of $\beta^T$, which subside after 80 generations.

**Table 5.** Details of design variables and their standard deviation values.

| Design Variable | Standard Deviation |
|---|---|
| $x_1$: Thickness of B-pillar inner | 0.03 |
| $x_2$: Thickness of B-pillar reinforcement | 0.03 |
| $x_3$: Thickness of floor side inner | 0.03 |
| $x_4$: Thickness of cross members | 0.03 |
| $x_5$: Thickness of door beam | 0.03 |
| $x_6$: Thickness of door beltline reinforcement | 0.03 |
| $x_7$: Thickness of roof rail | 0.03 |
| $x_8$: Material of B-pillar inner | 0.006 |
| $x_9$: Material of floor side inner | 0.006 |
| $x_{10}$: Barrier height | 10 |
| $x_{11}$: Barrier hitting position | 10 |

**Table 6.** The objectives and performance functions of Example 3.

| | |
|---|---|
| $f_1(\mu_{\mathbf{X}})$: | $1.98 + 4.9x_1 + 6.67x_2 + 6.98x_3 + 4.01x_4 + 1.78x_5 + 0.00001x_6 + 2.73x_7,$ |
| $f_2(\mu_{\mathbf{X}})$: | $(G_5(\mathbf{X}) + G_6(\mathbf{X}) + G_7(\mathbf{X}))/3,$ |
| $G_1(\mathbf{X})$: | $1.16 - 0.3717x_2x_4 - 0.00931x_2x_{10} - 0.484x_3x_9 + 0.01343x_6x_{10},$ |
| $G_2(\mathbf{X})$: | $0.261 - 0.01598x_1x_2 - 0.188x_1x_8 - 0.0198x_2x_7 + 0.0144x_3x_5 + 0.0008757x_5x_{10}$ $+ 0.08045x_6x_9 + 0.00139x_8x_{11} + 0.00001575x10x11$ |
| $G_3(\mathbf{X})$: | $0.214 + 0.00817x_5 - 0.1318x_1x_8 - 0.0704x_1x_9 + 0.030998x_2x_6 - 0.018x_2x_7 + 0.0208x_3x_8$ $+ 0.121x_3x_9 - 0.00364x_5x_6 + 0.0007715x_5x_{10} - 0.0005354x_6x10 + 0.00121x_8x_{11}$ $+ 0.00184x_9x_{10} - 0.018x_2^2$ |
| $G_4(\mathbf{X})$: | $0.74 - 0.61x_2 - 0.163x_3x_8 + 0.001232x_3x_{10} - 0.166x_7x_9 + 0.227x_2^2$ |
| $G_5(\mathbf{X})$: | $28.98 + 3.818x_3 - 4.2x_1x_2 + 0.0207x_5x_{10} + 6.63x_6x_9 - 7.77x_7x_8 + 0.32x_9x_{10}$ |
| $G_6(\mathbf{X})$: | $33.86 + 2.95x_3 + 0.1792x_{10} - 5.057x_1x_2 - 11.0x_2x_8 - 0.0215x_5x_{10} - 9.98x_7x_8 + 22x_8x_9$ |
| $G_7(\mathbf{X})$: | $46.36 - 9.9x_2 - 12.98x_1x_8 + 0.1107x_3x_{10}$ |
| $G_8(\mathbf{X})$: | $4.72 - 0.5x_4 - 0.19x_2x_3 - 0.01228x_4x_{10} + 0.009325x_6x10 + 0.000191x_{11}^2$ |
| $G_9(\mathbf{X})$: | $10.58 - 0.674x_1x_2 - 1.958x_2x_8 + 0.02054x_3x_{10} - 0.0198x_4x_{10} + 0.028x_6x_{10}$ |
| $G_{10}(\mathbf{X})$: | $16.45 - 0.489x_3x_7 - 0.843x_5x_6 + 0.0432x_9x_{10} - 0.0556x_9x_{11} - 0.000786x_{11}^2$ |

**Table 7.** Best, median, and worst HV values obtained via both methods, presented for car side impact example for different values of $\beta^T$. The best performances are highlighted in bold font.

| $\beta^T$ | SLMDE | DLMDE | $\beta^T$ | SLMDE | DLMDE | $\beta^T$ | SLMDE | DLMDE |
|---|---|---|---|---|---|---|---|---|
| | **0.8256** | 0.8216 | | **0.7175** | 0.7104 | | **0.5208** | 0.5200 |
| 1.0 | **0.8245**$^{+}$ | 0.8211 | 2.0 | **0.7137**$^{-}$ | 0.7095 | 3.0 | **0.5145**$^{=}$ | 0.5142 |
| | **0.8235** | 0.8196 | | **0.7078** | 0.7070 | | **0.4996** | 0.4912 |
| SD | $5 \times 10^{-4}$ | $\mathbf{4 \times 10^{-4}}$ | SD | **0.0021** | 0.0040 | SD | **0.0050** | 0.0082 |

**Table 8.** Number of function evaluations required by both methods for car side impact example.

| $\beta^T$ | SLMDE | DLMDE |
|---|---|---|
| 1.0 | 4,623,000 | $1.467 \times 10^9$ |
| 2.0 | 4,623,000 | $1.6970 \times 10^9$ |
| 3.0 | 4,623,000 | $1.7342 \times 10^9$ |

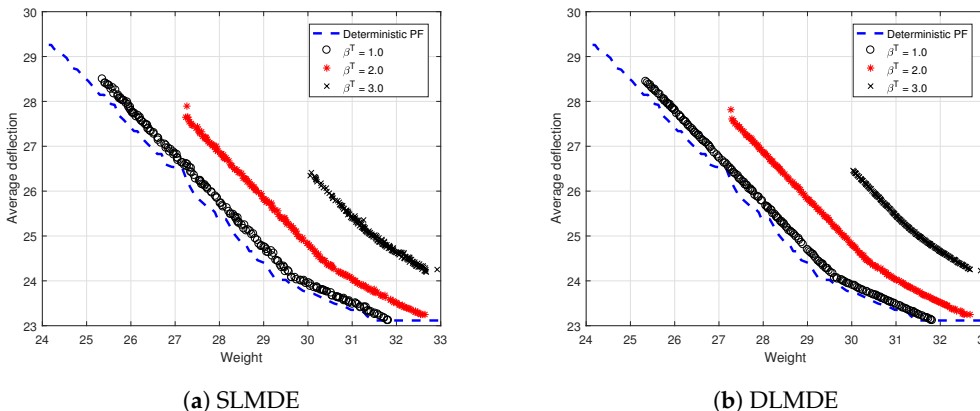

**(a)** SLMDE        **(b)** DLMDE

**Figure 6.** The obtained PO solutions by both methods for car side impact example for different $\beta^T$ values.

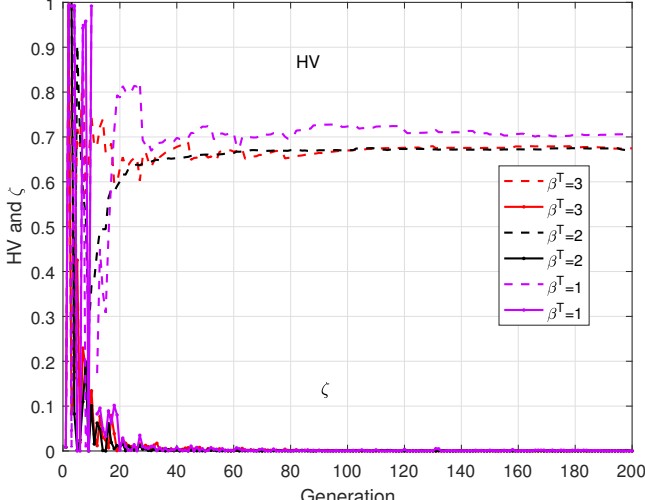

**Figure 7.** Progress of hypervolume and $\zeta$ of SLMDE with respect to number of generations for car side impact problem.

### 4.4. Example 4

The fourth example [44] consists of two objective functions and both of them are quadratic functions. The example has a linear performance function developed with three independent random normal variables. The RBDO formulation is given in Equation (24).

$$
\begin{aligned}
\text{min: } & f_1(\boldsymbol{\mu_X}) = (\mu_{x_1} - 1)^2 + (\mu_{x_2} - 2)^2 + (\mu_{x_3} - 3)^2, \\
\text{min: } & f_2(\boldsymbol{\mu_X}) = \mu_{x_1}^2 + 2\mu_{x_2}^2 + 3\mu_{x_3}^2, \\
\text{s.t.: } & P[G_i(\mathbf{X}) > 0] \leq \phi(-\beta_i^T), \quad i = 1, \\
& G_1(\mathbf{X}) = x_1 + x_2 + x_3 - 1, \\
& 0.1 \leq \mu_{x_i} \leq 6, \ i = 1, 2, 3.
\end{aligned}
\tag{24}
$$

where $x_1 \sim N(1, 0.05)$, $x_2 \sim N(2, 0.1)$, and $x_3 \sim N(3, 0.15)$.

Table 9 presents the statistical values of HV obtained via SLMDE and DLMDE. It can be observed that in most of the cases, SLMDE converged to better values of HV for different $\beta^T$. The HV values become reduced with larger values of $\beta^T$. The Wilcoxon test results are shown in the same table with symbols $(+, =, -)$. It can be seen from the table that SLMDE shows an equivalent performance with DLMDE for $\beta^T = 2$ and 3, and a bad performance for $\beta^T = 1$.

**Table 9.** Best, median, and worst HV values obtained by both methods are presented for example 4 for different values of $\beta^T$. The best performances are highlighted in bold font.

| $\beta^T$ | SLMDE | DLMDE | $\beta^T$ | SLMDE | DLMDE | $\beta^T$ | SLMDE | DLMDE |
|---|---|---|---|---|---|---|---|---|
| | 0.7733 | **0.7751** | | **0.7416** | 0.7413 | | **0.6834** | 0.6833 |
| 1.0 | **0.7723**$^-$ | 0.7721 | 2.0 | **0.7407**$^=$ | 0.7405 | 3.0 | 0.6826$^=$ | **0.6828** |
| | 0.7380 | **0.7585** | | **0.7297** | 0.7091 | | 0.6439 | **0.6799** |
| SD | 0.0100 | **0.0028** | SD | **0.0035** | 0.0059 | SD | 0.0103 | **0.0080** |

Figure 8 shows the reliable PO solutions generated in the run corresponding to the median HV value obtained via both methods for different values of $\beta^T$. As observed in the previous examples, for larger $\beta^T$, PO solutions move inside the feasible region, away from the deterministic PO front.

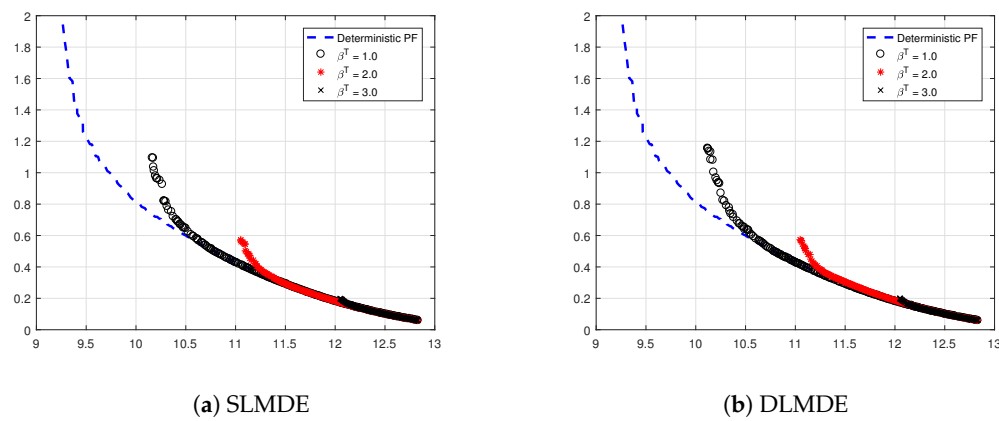

(**a**) SLMDE　　　　　　　　　　　　　　　　(**b**) DLMDE

**Figure 8.** The PO solutions obtained via both methods for example 4 for different $\beta^T$ values.

Table 10 presents the computational efficiencies of both methods. The proposed method requires only 50% of function evaluations as that of DLMDE. Figure 9 also shows similar observations for HV and $\zeta$ during the progress of the generations. There is an initial fluctuation which reduces after 20 generations.

**Table 10.** Number of function evaluations required by both methods for Example 4.

| $\beta^T$ | SLMDE | DLMDE |
|---|---|---|
| 1.0 | 280,000 | 520,000 |
| 2.0 | 280,000 | 520,000 |
| 3.0 | 280,000 | 520,000 |

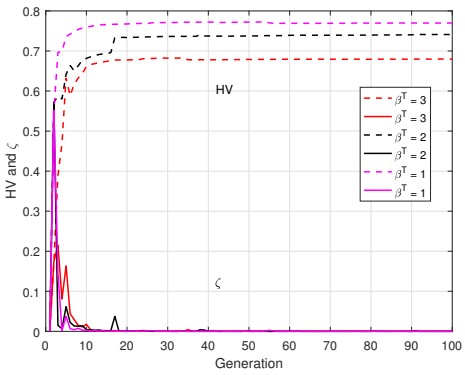

**Figure 9.** Progress of hypervolume and $\zeta$ of SLMDE with respect to number of generations for Example 4.

## 5. Conclusions

A single-loop multi-objective reliability-based design optimization has been proposed for generating reliable PO solutions quickly. It was developed by applying KKT optimality conditions to PMA for generating an approximate expression of MPTP. The search direction of approximate MPTP was modified via chaos control theory. The concept of the shifting vector approach was implemented with the novel formulation to include both target and trial vectors. DE was made adaptive, using the heuristic parameter that helped DE to perform different mutation operators. The proposed SLMDE was tested on three mathematical and one engineering bi-objective RBDO examples. It was found that SLMDE generated more reliable PO solutions for all examples compared to DLMDE. The results demonstrate that the convergence of SLMDE takes less function evaluations than DLMDE. For all four examples, the SLMDE was able to generate better HV values. For example 2, a lot of fluctuations during the progress of hypervolume can be observed, which stabilize gradually. The user-defined parameter $\zeta$ shows stable progress for all the examples. In the future, the proposed method can be modified for quick convergence by incorporating quantile approximation for reliability analysis. The proposed method can also be tested on other real-world examples having many nonlinear functions.

**Author Contributions:** Conceptualization: R.B. and D.S.; methodology: R.B.; validation: R.B.; formal analysis: R.B. and D.S.; investigation: R.B.; writing—original draft preparation: R.B.; writing—review and editing: R.B. and D.S.; visualization: R.B.; supervision: D.S. All authors have read and agreed to the published version of the manuscript.

**Funding:** This research received no external funding.

**Conflicts of Interest:** The authors declare no conflicts of interest.

## Abbreviations

The following abbreviations are used in this manuscript:

| | |
|---|---|
| MORBDO | Multi-objective reliability-based design optimization |
| MPTP | Most probable target point |
| SLCC | Single-loop chaos control |
| DE | Differential evolution |
| KKT | Karush-Kuhn Tucker optimality conditions |
| HV | Hypervolume performance indicator |
| SLMDE | Single-loop multi-objective differential evolution |
| DLMDE | Double-loop multi-objective differential evolution |
| PMA | Performance measure approach |

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
