# Peer review of "Single-Loop Multi-Objective Reliability-Based Design Optimization Using Chaos Control Theory and Shifting Vector with Differential Evolution"

_mca, doi:10.3390/mca28010026_

Round 1
Reviewer 1 Report
The paper presents the multi-objective reliability-based design optimization (MORBDO) for generating the Pareto solutions. Since the MORBDO method incurs expensive computational cost, the chaos control strategy is used to accelerate the computational efficiency and accuracy. In general, the topic is very important and it is more general than deterministic optimization. The design results reported therein are quite convincing and the manuscript is well-written. The reviewer therefore recommends publication of this paper with only a few minor comments as below
1. The expression of format should be unified, such as the capital and small letter. The Kriging should use the unified format.
2. The time-dependent reliability-based robust design optimization should be TRBDO other than RBRDO, which should be revised.
3. The authors use the MCC for reliability analysis because of the high performances. The authors should add the related good comment to explain the reason.
4. The Eq. (5) should be revised, the last formulation is AMV. It should be rewrite as a new formulation.
5. In introduction, the authors seem to be not aware of the recent state-of-the-art works on RBDO using metaheuristic algorithm, such as https://doi.org/10.1016/j.eswa.2022.117640.
6. Why authors use the differential evolution algorithm? There are many new metaheuristic algorithms with high performance. The related explanation should be added.
7. The Pareto solution is the real Pareto solution or approximate value? The validation are suggested to be added in the conclusion.
Reviewer 2 Report
A single-loop multi-objective reliability-based design optimization has been proposed for generating reliable PO solutions quickly in this manuscript. And the proposed method is tested by three examples. The logical structure of this manuscript is clear and the illustrations are concise and clear, but it needs to be improved in the following aspects. Please see the attachment.

Reviewer 3 Report
The authors propose an adaptation of a multi-objective differential algorithm to handle reliability problems. In this case, they propose to use chaos theory to avoid a double loop to compute the reliability. The authors show the application of their method to two academic problems as well as an application. The results show that it has advantages compared to the double loop approach in the experimental setting.
I find the paper interesting and it has its strongest contribution in avoiding the use of a double loop to estimate the reliability of multi-objective optimization problems as well as showing the method on an application.
However, in its current stage I find several areas of opportunity:
- The tests are performed only on low-dimensional problems
- The comparison is made only to the double loop approach and not with other strategies such as coevolution, external archivers, and surrogate models. Which have been used in the literature for similar problems
- The results are not investigated for statistical significance
- The advantages seen in terms of function evaluations could be made arbitrarily large by changing the evaluations in the nested loop
Specific comments
- Explanation of Equation 1, please add the domains of the sets and functions used
- The comparison is made with a standard algorithm with a double loop. However, there exist other approaches such as the use of coevolution, surrogate models, and external archivers. I would suggest contrasting the proposed approach to those methods in the text. Further, consider adding at least one to make the comparison on the examples. Some references are
- The examples shown are of low dimensionality. Examples 1 and 6 have two and six variables respectively. It would be interesting to see how the method scale with the number of dimensions
- In Tables 1 and 3, please add the standard deviation of the results
- In Tables 2 and 4, the number of function evaluations of DLMDE is by design larger. It would be interesting to either put a target for the hypervolume computation or compare the number of function evaluations required to get to that target.
- The results show the hypervolume computation. However, it would be interesting to see as well the constraint violation to compare it among the algorithms as well as the failure probabilities since the main goal of the approach is to have reliable results.
- Further, it would be interesting to investigate the reliability of the nominal fronts under the proposed scenarios to show the advantages of looking for reliable solutions despite the extra computational resources used
- Please include the reference point used for the hypervolume calculation
- Please investigate all the results for statistical significance using a non-parametric test
Some references of approaches that avoid the double loop for related problems (note that many other approaches exist)
- https://arxiv.org/abs/1303.3901
-https://asmedigitalcollection.asme.org/mechanicaldesign/article/135/2/021006/375754/Co-Evolutionary-Optimization-for-Multi-Objective
- https://ieeexplore.ieee.org/document/5949877
Reviewer 4 Report
The paper is well organized and the language used is clear, allowing one to understand the complex concepts presented. The problem to be solved is clearly justified taking into account the limitations of the literature. Also, the details of the formulation and of implementation were given in detail and in a clear way.
Given this, I only suggest that the methodology developed be applied to more test problems.
Round 2
Reviewer 3 Report
The authors have completed the review. In my opinion, the paper is ready for publication.